# Concussion Disrupts Sleep in Adolescents: A Systematic Review and Meta-Analysis

**DOI:** 10.3390/clockssleep7030046

**Published:** 2025-08-29

**Authors:** Suleyman Noordeen, Poh Wang, Anna E. Strazda, Eszter Sara Arany, Mehmet Ergisi, Linghui Janice Yeo, Rebeka Popovic, Abinayan Mahendran, Mikail Khawaja, Kritika Grover, Mohammed Memon, Saahil Hegde, Connor S. Qiu, Charles Oulton, Yizhou Yu

**Affiliations:** 1International Sleep Charity, Shedfield, Southampton SO32 2HQ, UK; shin2@cam.ac.uk (S.N.); pjw112@ic.ac.uk (P.W.); aes99@cam.ac.uk (A.E.S.); mehmetergisi@gmail.com (M.E.); rebeka.popovic@ucl.ac.uk (R.P.); abinayan.mahendran17@imperial.ac.uk (A.M.); mk2079@cam.ac.uk (M.K.); mohammed.memon008@gmail.com (M.M.); saahil0987@gmail.com (S.H.);; 2School of Clinical Medicine, Cambridge Biomedical Campus, University of Cambridge, Box 111, Cambridge CB2 0SP, UK; 3School of Public Health, Faculty of Medicine, Imperial College London, Exhibition Road, South Kensington, London SW7 2AZ, UK; 4School of Medicine, Imperial College London, London SW7 2AZ, UK; 5Department of Medicine, Norfolk and Norwich University Hospital, Norfolk and Norwich University Hospitals NHS Foundation Trust, Colney Ln, Norwich NR4 7UY, UK; 6Department of Medicine, Leeds Teaching Hospital NHS Trust, St James University Hospital, Beckett St, Leeds LS9 7TF, UK; 7Department of Medicine, Eastbourne District General Hospital, East Sussex NHS Trust, Eastbourne BN21 2UD, UK; 8Healthspan Biotics, Milner Therapeutics Institute, Jeffrey Cheah Biomedical Centre, Puddicombe Way, Cambridge CB2 0AW, UK

**Keywords:** concussion, sleep-related symptoms, systematic review, meta-analysis, Pittsburgh Sleep Quality Index

## Abstract

Concussions significantly impact sleep quality among adolescents. Despite increasing recognition of these effects, the complex relationship between adolescent concussions and sleep disturbances is still not fully understood and presents mixed findings. Here, we conducted a systematic review and meta-analysis to assess how concussions affect sleep-related symptoms in adolescents. We included papers presenting Pittsburgh Sleep Quality Index (PSQI) differences following concussion in high school and collegiate patients, with sleep measures recorded following concussion. Animal studies, research on participants with conditions other than concussion, non-English articles and papers failing to present PSQI data were excluded. We searched MEDLINE^®^, Embase^®^, CINAHL, Web of Science™, PsycINFO^®^, Google Scholar, and Cochrane Central Register of Controlled Trials databases until 23 August 2024. In addition, we performed hand-searching of relevant reference lists and conference proceedings to identify further studies. Risk of bias was assessed using the Newcastle–Ottawa scale. In total, we considered 4477 studies, with nine meeting the inclusion criteria. Our analysis involved 796 participants, 340 of whom had experienced a concussion. Analysis was carried out using the meta and metafor packages in R (version 4.0.0). We showed a deterioration in sleep quality post-concussion, as evidenced by increased PSQI (standardised mean difference 0.84; 95% CI 0.53–1.16; *p* < 0.0001). Subgroup and quality assessments confirmed the consistency of these findings. Since poor sleep quality impacts daytime activities, we analysed the relationship between concussion and daytime dysfunction. We showed that concussion is associated with a significant worsening of the daytime dysfunction score by 0.55 (95% CI 0.24–0.70; *p* = 0.006). We conclude that concussions impair both sleep quality and daytime functioning in adolescents. Our research underscores the need for systematic inclusion of sleep quality assessments in post-concussion protocols and calls for targeted interventions to manage sleep disturbances post-concussion to mitigate their broader impacts on daily functioning.

## 1. Introduction

Poor sleep is prevalent across different demographic groups [1,2] and can negatively impact cognitive performance, mental health, and quality of life [3]. Head injuries, including concussion, can disrupt sleep [4], with over 40% of participants experiencing sleep difficulties after a head injury [5]. Concussion is a type of traumatic brain injury induced through biomechanical forces [6] and accounts for 15% of injuries reported in high school sports over a school year, though it may be lower than reality due to underreporting [7,8]. There is a growing body of evidence focusing on adolescent concussion; however, a large bulk of paediatric studies focus on accident-related concussions [7,9,10,11]. Adolescents and collegiate-aged students are also uniquely susceptible to sleep-related complications, given the importance of sleep in neurological development, and as such, could have different responses from adults with a concussion [12]. Concussions have multiple adverse effects and feature a diversity of symptoms, including vertigo, fatigue, sleep disturbances and cognitive impairment [6,13], which are underpinned by several biological markers, including astrocytic dysregulation, blood–brain barrier malfunction, tau-protein deposition, as well as white and grey matter changes [14,15,16,17].

Concussions in adolescent and collegiate athletes can cause long-term consequences on neurological development [18]. Of the symptoms of concussion, sleep disturbance is particularly relevant, as disrupted sleep during adolescence is associated with long-lasting impacts on brain development [19]. A concussion can affect sleep quality via several mechanisms. Circadian rhythm disruption post-concussion, possibly due to aberrant melatonin release, could be a major contributor to sleep disturbance [20]. Similarly, astrocytic dysregulation could compromise the regulation of sleep, as normal neuronal activity during sleep relies on glymphatic activation and lactate provision regulated by astrocytic calcium signalling [21]. Animal models of concussion showed that restoring astrocyte function by manipulating adenosine signalling alleviates cognitive impairment [22]. The sleep-concussion relationship is bidirectional, with poor sleep lengthening time for recovery post-concussion and contributing to further impairments in executive function [23,24,25].

Despite the importance and scale of concussion-related sleep disruption, the effect size of concussion on sleep quality remains mixed. The major limitations to research in the field derive from inconsistent measures used to evaluate sleep disturbance, with techniques ranging from biometric data to a variety of questionnaires, many of which only partially evaluate sleep characteristics [26].

Some clinical evaluations of post-concussion symptomatology, such as the post-concussion symptom scale (PCSS), include measures on sleep symptoms [19], while others, like the immediate post-concussion assessment and cognitive testing (ImPACT) [27], do not. A scoping review evaluating sleep disturbances after sport-related concussion in adults finds evidence of sleep disturbance after concussion but concludes that the effect size of sleep disruption after concussion is unclear, with the variety of methods used in different studies resulting in heterogeneity of outcomes [28]. In line with this, whilst some research shows that concussion is linked to worse sleep quality, others do not replicate this [25,28]. Our pilot search [29] found that of the various sleep-symptom-specific questionnaires used, only the Pittsburgh Sleep Quality Index (PSQI) and Epworth Sleepiness Scale (ESS) [30] were used frequently enough to allow meaningful synthesis. The PSQI [31] is a validated self-report questionnaire that scores multiple components of sleep: sleep quality, latency, duration, efficiency, sleep medication, and daytime functioning. This questionnaire provides a unified measure to consistently assess sleep following concussion, as well as providing further resolution as to which characteristics are most prominently affected and estimating an effect size [32].

Importantly, reliable estimates of concussion-related sleep disruption in adolescents are urgently needed to inform treatment efforts and mental health-related measures. To address this gap, our study synthesises the literature on concussion in adolescent and college-aged populations and conducts a meta-analysis using the PSQI to clarify the overall impact of adolescent concussion on sleep, as well as identify which components of sleep are affected. By summarising data from nine studies involving 796 participants, we identified a deterioration in sleep quality and daytime dysfunction.

## 2. Materials and Methods

This systematic review was developed following the Preferred Reporting Items for Systematic Reviews and Meta-Analyses guidelines [33]. The search methodology was registered with PROSPERO (CRD42021290208).

### 2.1. Eligibility Criteria

We included all RCTs, abstracts, cross-sectional, and case–control studies published between 1988 and August 2024 that evaluated the impact of concussion on sleep quality expressed in PSQI. Our inclusion criteria were: 1. Sleep measures recorded within two years of concussion; 2. People with a concussion or mild traumatic brain injury; 3. University or college students and adolescents. This was defined with an age cutoff of 12–30 years old, to account for older collegiate athletes.

The exclusion criteria were as follows: 1. animal studies; 2. participants having neurological conditions other than concussion or mild traumatic brain injury; 3. articles not in the English language; 4. cohort or article duplicates; 5. articles that did not have the full text or results.

### 2.2. Literature Search Strategy

We conducted a systematic review of all articles in the MEDLINE^®^, Embase^®^, CINAHL, Web of Science™, PsycINFO^®^, Google Scholar, and Cochrane Central Register of Controlled Trials databases until 23 August 2024. In addition, we performed hand-searching of relevant reference lists and conference proceedings to identify further studies. Pre-registration was performed in accordance with best practice guidelines [34,35]. In line with these practices, we conducted an initial scoping and pilot study on the links between concussion and poor sleep [29]. Our original scoping review [29] identified articles that used other sleep questionnaires to measure sleep; however, these failed to reach a sufficient number for analysis. Our search strategy was centred on keywords and medical subject headings (MeSH) associated with ‘sleep’ and ‘concussion’.

The searching algorithms included “concuss*” and “sleep”. We provided the detailed search phrases on the Prospero records (https://www.crd.york.ac.uk/PROSPEROFILES/245605_STRATEGY_20210407.pdf) (accessed 30 January 2025). A full database of the identified studies is available in the Appendix A, available at https://github.com/izu0421/sleepConcussion (accessed 30 January 2025). We followed the PRIMSA (Meta-analysis Of Observational Studies in Epidemiology) guidelines for the reporting and methodology of our meta-analysis [34], available here: https://github.com/izu0421/sleepConcussion (accessed 30 January 2025).

### 2.3. Main Outcomes

The outcomes searched included sleep duration in hours, measurement of insomnia based on the Pittsburgh Sleep Quality Index (PSQI) and measurement of daytime sleepiness using the Epworth Sleep Scale. There were only sufficient studies with the PSQI as an outcome variable for quantitative meta-analysis, and the Epworth Sleep Scale was excluded from the quantitative analysis. The other markers were discussed qualitatively. The PSQI score can range from 0 to 21, where lower scores denote a healthier sleep quality [31].

### 2.4. Data Extraction and Curation

All authors took part in the data extraction and curation process. The filtering process was performed in accordance with PRISMA guidelines [33]. Reviewers applied eligibility criteria and selected studies for inclusion and data extraction. Due to the large number of articles obtained, there were two stages of study selection to fulfil our selection criteria for data extraction. First, two reviewers filtered the titles and abstracts of each study identified by the search. The reviewers were not blinded to each other’s decisions. The study was included in the next step when both reviewers agreed on its inclusion. In the case of a disagreement, a third reviewer weighed in on the decision to include the study based on an assessment of the entire publication. In the second step, two independent reviewers examined the full publication of the paper to decide on its inclusion. We provide the study selection process in the Appendix A and also describe it in Figure 1. The data extracted included the study size, gender, sleep duration, PSQI, and ESS. Two authors independently extracted the data. In the case of missing data, the authors of the study were contacted.

### 2.5. Quality Assessment

Study quality was assessed using the Newcastle–Ottawa [36] quality assessment scale for case–control studies, a validated scale that is well-suited to systematic reviews and is particularly well-suited for psychiatry [37]. This tool evaluates studies on eight different domains (adequate case definition, representativeness of the cases, selection of controls, definition of controls, comparability of cases and controls, ascertainment of exposure, same method of ascertainment, non-response rate), with a number output which can then be used to perform moderator analysis. This output is a nine-star scale. An overall summative assessment was made for each study and is available on our GitHub repository: https://github.com/izu0421/sleepConcussion (accessed 30 January 2025). Risk of bias is discussed as high, medium, or low bias in line with the NOS criteria [36].

### 2.6. Data Analysis

The meta-analysis was conducted using the Meta and Metafor R packages in R [38,39,40] following best practices. Random-effects models (DerSimonian and Laird method) and mixed-effect models were used, as this accounts for heterogeneity. We reported the weighted standardised mean difference and its respective 95% confidence intervals (CIs). Heterogeneity was assessed using the I2 statistic [41]. Publication bias was assessed visually via funnel plots [42]. All analyses, source data, and additional tests to fully reproduce the results are found in our GitHub repository: https://github.com/izu0421/sleepConcussion and presented as a webpage in detail on https://www.yizhouyu.com/sleepConcussion/ (accessed 30 January 2025).

## 3. Results

### 3.1. Study Inclusion and Filtering

The initial search yielded 4477 articles (see Figure 1 for workflow). Of these articles, 2791 were duplicates and were removed, leaving 1686 titles and abstracts, which were screened for relevance. A total of 1139 articles were removed by screening the title and abstract (Filter 1), leaving 547 full-text articles to be assessed for eligibility by evaluating the full text (Filter 2). A total of 470 full-text articles were excluded in Filter 2, leaving 77 studies that were relevant. Of these, 68 were excluded in the Filter 3 process by selecting for studies that had PSQI measurement on both concussed and non-concussed individuals, leaving 9 studies (Table 1). The date of publication ranged from 2008 to 2023. Of these, two were abstracts [43,44]. Three studies used a case–control approach [9,44,45], of which one was an abstract [44]. Four studies were retrospective cohort trials or longitudinal [46,47,48,49], with the data overall including 796 participants, of which 340 were concussed.

### 3.2. Study Setting

Nine studies were analysed (see Table 1). Seven studies were performed in the United States [9,43,44,46,49,50]. One study was performed in Australia [49], and one in Canada [45]. Patients recruited consisted of college and high school athletes, as well as university students, with an average age of 18.6 years. The studies included 408 males and 388 females, corresponding to a sex ratio of 1.05:1 (male/female).

Recruitment was carried out by team physicians, university healthcare systems, and sports clinics. One study was focused solely on football players [43], whilst four other studies evaluated both contact and non-contact sports [44,45,46,47], and four others did not solely focus on sports leading to injury [9,48,49,50]. The control groups were drawn from a similar student population in eight of the studies, with one study using an additional separate clinical population suffering from orthopaedic injury as a second comparator group; however, the PSQI mean and range for this group were similar to the student population control groups (PSQI = 2.52, standard deviation: 3.48) [50]. One study did not provide a breakdown of patients by mode of injury [50].

### 3.3. Meta-Analysis on the Effect of Concussion on Sleep Quality

The nine prospective longitudinal cohort studies that fulfilled the systematic review criteria consisted of 340 concussed participants and 456 controls. The mean age of the concussed participants across studies was 18.57 years (standard deviation: 2.20), which is similar to that of the concussed individuals (18.62, standard deviation: 2.63). Across the studies, the percentages of female participants were 44.04% (standard deviation: 21.6%) in controls and 43.90% (standard deviation: 23.1%) in concussed participants. Given the consistency of the cohort parameters between the concussed and control groups, we next analysed the standardised mean difference in the sleep quality between the control and concussed participants. We found a significant increase in the PSQI of 0.84 (95% CI 0.53–1.16) in participants with a concussion (*p* < 0.0001), which corresponds to a 4.24% decrease in sleep quality (Figure 2).

Individually, three of the included studies found a non-statistically significant difference in overall PSQI score between the concussed and non-concussed participants [43,48,49]. One of these, however, found a significant increase in the daytime dysfunction subcategory of the PSQI [49]. Similarly, Hoffman and colleagues found significant worsening in sleep efficiency [9]. Gosselin et al. found significant differences in daytime dysfunction (*p* < 0.05), sleep quality (*p* = 0.01), and sleep disturbances (*p* < 0.05) [45]. Smulligan and colleagues [46] showed that concussed individuals had worse PSQI scores than the control group. Similarly, Schmidt and colleagues [50] showed that concussions negatively affected sleep quality, indicated by increased PSQI scores, which were still present after 3 months. Five studies only provided data on global PSQI scores [43,44,46,47,48]. Smulligan and colleagues found that there was no difference in sleep quality and single-task tandem gait between concussed individuals who participated in early physical activity and that of the control group, whereas the concussed individuals without early physical activity reported significantly worse [47].

### 3.4. Risk of Bias, Heterogeneity, and Confounder Analyses

Given the presence of heterogeneity in our primary analysis, we investigated which study might have contributed to this through a funnel plot. Albicini and colleagues [49] reported an effect size that was most distinct from others (Figure 3a). The presence of potential heterogeneity indicates that the overall standardised mean difference is better estimated through a random effect model compared to a common effect model (Figure 2).

We next looked at possible variables that could affect our estimation of the effect of concussion on sleep quality. We found no moderating effect for sex (Figure 3b) on PSQI, agreeing with existing literature findings of no difference between sexes and PSQI score in concussion [51]. We also found that the average age of the study cohort did not impact PSQI (Figure 3c). Since two of the nine analysed studies were not subject to peer review, we asked whether the quality of a study could be a potential confounder in our meta-analysis. We calculated the Newcastle–Ottawa scale for case–control studies of each study to quantify their quality [52]. The range of scores generated ranged from 2 to 8 out of 9, with higher scores indicating a lower risk of bias. One abstract included in our study scored four, largely due to a lack of data presented [43]. Of the papers included, one scored a seven [50] and five scored an eight [9,45,46,47,48]. The other abstract scored an eight [44]. One of the papers scored a two [49], due to a lack of detail on study participant characteristics, though sleep and concussion were not the main focus of this paper. We found that the quality of the papers used in our research did not influence our estimation of the effect of concussion on sleep quality (Figure 3d). Lack of full data presentation was the most common category responsible for increasing the risk of bias. Three studies were drawn from an adolescent population aged 12–18 years, potentially introducing heterogeneity when compared to the student population [46,47,48]. Taken together, despite the presence of study-specific variations and the inherent heterogeneity in the data, our meta-analysis robustly supports the impact of concussions on sleep quality in adolescent athletes.

### 3.5. Concussion Impacts Daytime Dysfunction

The daytime dysfunction component of the PSQI clinically manifests as impaired wakefulness and reduced daily productivity, constitutes a significant metric of overall sleep quality, and has substantial implications for cognitive and behavioural performance [31,52,53]. It compromises academic and social functioning and may precipitate broader mental health issues [54]. Since poor sleep quality increases the risk of daytime dysfunction, and our results showed that concussion significantly worsens sleep quality, we next investigated the relationship between concussion and daytime dysfunction. Our analysis comprised three studies with 404 participants (35.6% concussed). We found that concussion is associated with a significant increase in the daytime dysfunction score, with a standardised effect size of 0.55 (95% CI 0.24–0.70; *p* = 0.006, Figure 4a), which corresponds to an approximate 9.2% increase in the daytime dysfunction score. Our analysis shows no evidence of heterogeneity (Figure 4a,b) or bias related to gender, age, or the Newcastle–Ottawa scale of the study (Figure 4c–e). Taken together, our results indicate that a concussion significantly worsens the daytime dysfunction.

## 4. Discussion

Our meta-analysis investigated the hypothesis that concussion in adolescents resulted in poorer sleep outcomes as evaluated with the PSQ [31]. Sleep disruptions following concussion are being increasingly recognised as a major factor in determining concussion sequelae, with sleep symptoms having associations with impaired post-concussion executive function recovery, as well as influencing mental health and physical recovery [25,55]. Our study provides robust evidence that concussions significantly impair sleep quality and daytime functioning in adolescents. Across nine studies involving 796 participants, we found that concussed adolescents consistently reported poorer sleep quality, as reflected by elevated PSQI scores, and greater daytime dysfunction compared to non-concussed peers. The magnitude of the association, coupled with the consistency across studies, highlights that sleep disturbances are not merely secondary or transient symptoms but likely represent a core component of the post-concussion syndrome in young athletes.

Several studies reported contrasting results on the association between concussion and sleep quality. Bone and colleagues found a significantly reduced time asleep based on actigraphy data, whereas Hoffman and colleagues found no difference. Instead, Hoffman and colleagues reported an increased sleep-onset latency, but a different study found no differences in sleep latency via polysomnography [45]. In addition to actinography data, two studies included the Epworth Sleepiness Scale alongside the PSQI [9,45]. No significant difference between groups was identified in the study by Gosselin and colleagues, whereas a significantly higher ESS was reported for the concussed group in the study Hoffman and colleagues [9].

Our results concur with previous literature evaluating the effects of concussion on sleep in adult athletes, indicating common mechanisms in concussion for both adolescent and adult patients, as well as replicating the findings in our pilot investigation [28,29]. Additionally, this outcome corresponds with sleep-related consequences due to concussion from specific aetiologies, such as blast-related or motor-vehicle accident-related concussion [9,10,56]. Our focused analysis identifies that a concussion significantly impacts daytime dysfunction. In line with this, the daytime dysfunction component of the PSQI was the most consistently reported altered component, even in cases where no overall difference in PSQI was found [48,49]. This suggests that while overall sleep quality as measured by the PSQI may not always appear to deteriorate following a concussion, specific aspects such as sleep efficiency and the ability to function during the day are consistently affected. The persistent disruption in these domains underscores the potential for concussions to cause subtle yet significant alterations in sleep architecture that standard sleep metrics like the PSQI might not fully capture. To better understand these effects, integrating neuroimaging, polysomnography, and physiological sleep assessments could provide deeper insights into the specific brain mechanisms affected by concussions.

Our results have several implications for the wider healthcare community, given the relevance of sleep to brain health [57,58,59]. Several long-term imaging studies have demonstrated structural changes in the brain in response to concussion, including altered neurite density in sleep-related white matter tracts [48,60]. Adolescent and collegiate athletes are at risk of repeat concussion, which is associated with poorer performance on neurocognitive testing compared to those with a single concussion [61,62]. Importantly, it is worth noting that the participants in the studies included in our analysis have a balanced number of each sex and are of similar age. Our analysis of potential confounding variables, including average age and sex ratio of each study, also showed no associations between these variables and sleep quality. All studies were recruited from English-speaking countries. This indicates that the population in our analysis is homogeneous. Thus, our characterisation and summary of the effect of concussion on PSQI and daytime dysfunction serves as a blueprint for further research to compare against.

Our study has a few limitations. Since concussions could have sexually dimorphic outcomes, with worse outcomes for female patients [62,63], future research can further explore the potential impacts of concussions across diverse populations to better understand the influence of age, sex, and ethnic differences on concussion outcomes. We were also unable to perform moderator analysis for specific confounding factors, such as mental health differences or differences in sleep hygiene awareness, both of which influence sleep, though some of the included studies performed moderator analysis to account for this [45,49,55,61]. Investigation of longitudinal outcomes and their relationships with early sleep disruption would inform clinician management and return-to-play decision-making, with the aim of avoiding long-term developmental consequences [64,65].

Overall, our systematic analysis of existing work on concussion in adolescents and sleep quality demonstrates the need for sleep clinicians and sports staff to work together to minimise the risks posed by concussion.

## Figures and Tables

**Figure 1 clockssleep-07-00046-f001:**
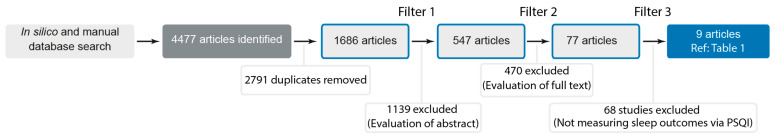
Data filtering methodology flowchart. The PRISMA (Preferred Reporting Items for Systematic reviews and Meta-Analyses 2020) workflow was followed.

**Figure 2 clockssleep-07-00046-f002:**
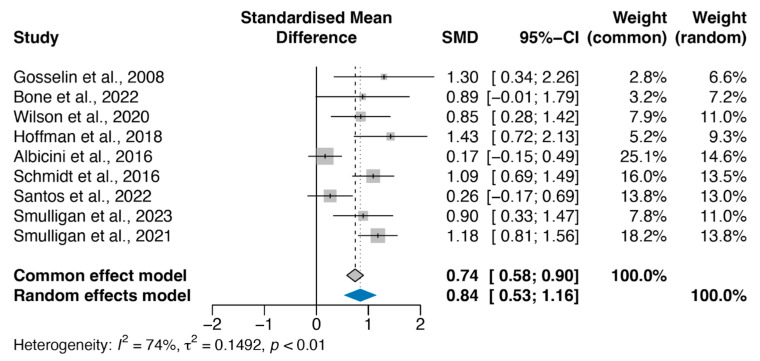
Adolescent sleep quality is significantly worse after concussion events. The individual and pooled standardised mean differences (abbreviated as SMD in the figure) are shown. The pooled SMD was a PSQI of 0.84 (95% CI 0.53–1.16; *p* < 0.0001) in adolescents who sustained a concussion compared to controls (random effect model). The size of the box representing the point estimate for each study in the forest plot is proportional to the contributing weight of that study estimate to the summary estimate. There is a presence of considerable heterogeneity as indicated by the 74% *I*^2^ score and *p* < 0.01 [9,43,44,45,46,47,48,49,50].

**Figure 3 clockssleep-07-00046-f003:**
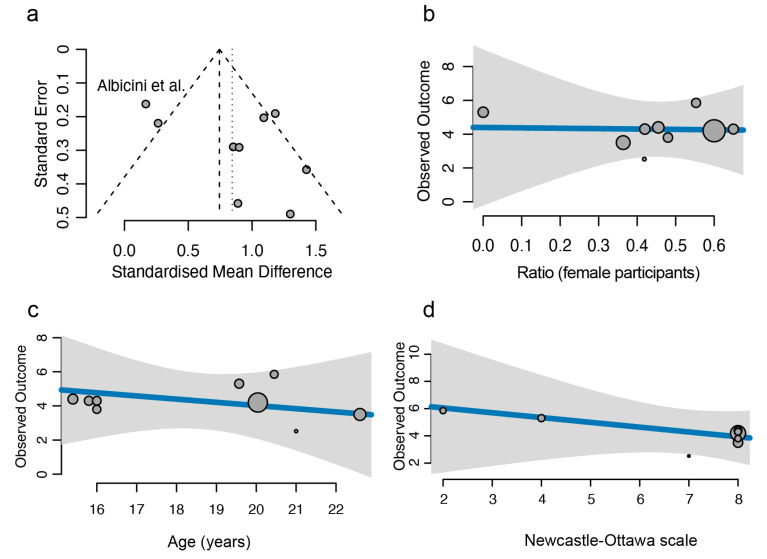
Publication bias and confounder analysis. (**a**) Funnel plot indicating the possible heterogeneity in the analysis. The standard error in the y-axis is a measure of study precision, and the standardised mean difference in the x-axis indicates the study results. Each dot represents an individual study, and the outlier is labelled. The outer dashed lines indicate the triangular region within which 95% of studies are expected to lie in the absence of both biases and heterogeneity. The middle dashed line indicates the overall standardised effect size determined via a common effect model, and the vertical dotted line represents the overall standardised effect size determined via a random effect model. (**b**–**d**) Confounder analysis of gender ratio (**b**), age (**c**), and Newcastle–Ottawa scores (**d**). The gender ratio, average age, and Newcastle–Ottawa scores of each study are plotted against the observed outcome and modelled in a meta-regression analysis. The size of each dot represents the weight of each study. All *p*-values of these confounders are above 0.05 [49].

**Figure 4 clockssleep-07-00046-f004:**
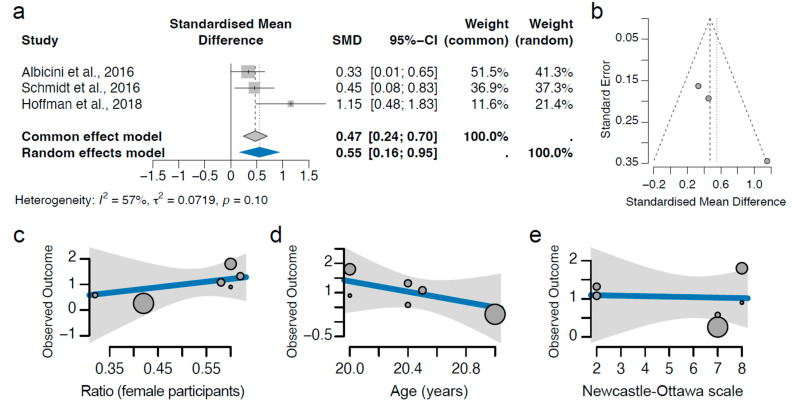
**Relationship between concussion and daytime dysfunction**. (**a**) The individual and pooled standardised mean difference (abbreviated as SMD in the figure) are shown. The pooled SMD was 0.55 (95% CI 0.24–0.70; *p* = 0.006) in adolescents who sustained a concussion compared to controls (random effect model). (**b**) Heterogeneity analysis. The standard error in the y-axis is a measure of study precision, and the standardised mean difference in the x-axis indicates the study results. Each dot represents an individual study. (**c**–**e**) Confounder analysis of gender ratio (**c**), age (**d**), and Newcastle–Ottawa scores (**e**). The gender ratio, average age, and Newcastle–Ottawa scores of each study are plotted against the observed outcome and modelled in a meta-regression analysis. The size of each dot represents the weight of each study. All *p*-values of these confounders are above 0.05 [9,49,50].

**Table 1 clockssleep-07-00046-t001:** **Study characteristics**.

Reference	Authors	Title	Year	Total	Non-Concussed	Concussed
[9]	Hoffman, N. L., O’Connor, P. J., Schmidt, M. D., Lynall, R. C., & Schmidt, J. D	“Differences in sleep between concussed and non concussed college students: a matched case–control study.”	2019	40	20	20
[43]	Bone, T., Konz, S. M., Garrett, W., & Gilliland, C. A.	“The Effects of Concussion on Quantity and Quality of Sleep in Football Athletes.” *	2022	27	20	7
[44]	Wilson, J. C., Kirkwood, M. W., Provance, A. J., Walker, G. A., Wilson, P. E., & Howell, D. R.	“Adolescents report sleep quality impairments acutely postconcussion.” *	2020	68	51	17
[45]	Gosselin, N., Lassonde, M., Petit, D., Leclerc, S., Mongrain, V., Collie, A., & Montplaisir, J.	“Sleep following sport-related concussions.”	2008	21	11	10
[46]	Smulligan KL, Wilson JC, Seehusen CN, Wingerson MJ, Magliato SN, Howell DR	“Post-Concussion Dizziness, Sleep Quality, and Postural Instability: A Cross-Sectional Investigation”	2021	131	73	58
[47]	Smulligan KL, Wingerson MJ, Little CC, Wilson JC, Howell DR.	“Early physical activity after concussion is associated with sleep quality but not dizziness among adolescent athletes”	2023	55	21	34
[48]	Lima Santos JP, Kontos AP, Holland CL, Stiffler RS, Bitzer HB, Caviston K, Shaffer M, Suss SJ Jr, Martinez L, Manelis A, Iyengar S, Brent D, Ladouceur CD, Collins MW, Phillips ML, Versace A	“The role of sleep quality on white matter integrity and concussion symptom severity in adolescents”	2022	90	33	57
[49]	Albicini, M. S., Lee, J., & McKinlay, A	“Ongoing daytime behavioural problems in university students following childhood mild traumatic brain injury”	2016	244	197	47
[50]	Schmidt A. T., Li X., Hanten G. R., McCauley S. R., Faber J. & Levin H. S.	“A Longitudinal Investigation of Sleep Quality in Adolescents and Young Adults After Mild Traumatic Brain Injury”	2016	43	77	
**Total:**				796	456	340

* The non-peer-reviewed studies are labelled with an asterisk.

## Data Availability

All analyses, source code and additional tests to fully reproduce the results are found in our GitHub repository: https://github.com/izu0421/sleepConcussion and presented as a webpage in detail on https://www.yizhouyu.com/sleepConcussion/. All necessary data to fully reproduce the results are found in our GitHub repository: https://github.com/izu0421/sleepConcussion. Additional data supporting the findings of this study are available from the corresponding author upon reasonable request.

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
