# Peer review of "Concussion Disrupts Sleep in Adolescents: A Systematic Review and Meta-Analysis"

_2624-5175, 2025, doi:10.3390/clockssleep7030046_

Round 1

Reviewer 1 Report

Comments and Suggestions for Authors

Dear Author(s),

Your manuscript is very interesting and comprehensively written. However, there are some improvement suggestions I would highly recommend to to take into consideration. 

Kindly, see below:

  • please move all section 4 to be section 2 after the Introduction and renumber the sections
  • it would be useful to emphasize in the introduction why the study is important
  • write a conclusion for the study with the main results
  • in the discussion section please include some limitations of the study, maybe one limitation can be the selection of only the databases etc. 

TThank you!

Author Response

We thank the reviewer and agree with all their suggestions. Please find attached the point-by-point reply.

Reviewer 2 Report

Comments and Suggestions for Authors

This is a good manuscript regarding the effects of concussion on adolescent sleep. It is not an article about sports-related concussion. The title and multiple statements are misleading. Only 5 of the nine studies pertained to sports-related concussion. The title should be Concussion disrupts sleep in adolescents. The first line of the abstract is misleading and must be changed. Line 27 states that you conducted a study of the effects of sports-related concussion. You did not. Line 29 states that you included papers presenting differences following sports-related concussion. That is not true. Line 31 states that conditions other than sports-related concussion were excluded. That is false. Line 45 mentions a conclusion about adolescent athletes. That is not justified by the data. Please revise the paper to correct all the misleading statements.

Author Response

(The authors gave the same response as above.)
